# Applying Level of Detail in a BIM-Based Project: An Overall Process for Lean Design Management

**Petteri Uusitalo [1],\*, Olli Seppänen [1], Eelon Lappalainen [1], Antti Peltokorpi [2]**  **and Hylton Olivieri [2]**

[1] Department of Civil Engineering, Aalto University, FIN-02130 Espoo, Finland; olli.seppanen@aalto.fi (O.S.); eelon.lappalainen@aalto.fi (E.L.)

[2] Racional Engenharia Ltda, São Paulo 04551-065, Brazil; antti.peltokorpi@aalto.fi (A.P.); hylton.olivieri@gmail.com (H.O.)

\* Correspondence: petteri.uusitalo@aalto.fi

**Abstract:** Few construction companies apply the available lean tools and processes in an integrated manner when managing design. Additionally, lean design management tools and processes each have their own strengths and optimal phases in which they should be applied. Earlier approaches in lean design management have not explicitly included the level of detail of building information model (BIM) models in connection with planning methods. For example, the Last Planner System (LPS) uses collaborative social methods to obtain task dependencies and commitments from project stakeholders, but it does not provide any guidance regarding what those tasks should be in a BIM-based process. With regard to production, this guidance is provided by combining location-based methods, such as a location-based management system or takt planning, to LPS. In a similar manner, by combining information from various sources, this paper defines a location-based design management process using the concept of level of detail that can be integrated with LPS. The level of detail definition must be based on the requirements of the end-user in each stage of the construction project. The process was cocreated and validated in focus group meetings with design and construction companies and further evaluated and assessed by applying it in a case project.

**Keywords:** level of detail; lean design management; BIM; last planner system

---

## 1. Introduction

Lean design management (LDM) has contributed to achieving remarkable results in construction projects, such as increasing collaboration among project parties, adding customer value, and reducing project costs (e.g., [1–3]) as well as solving design management problems [4]. LDM tools and processes each have their own strengths and optimal phases when they should be applied in a project [5]. For example, target value design (TVD) has yielded excellent results in controlling project costs and customer value when used in the project definition phase [6]. However, few construction companies apply the available lean tools and processes in an integrated manner. These companies could benefit from considerable synergies during the project lifecycle, such as better-aligned design and construction schedules, by varying and combining different lean processes and tools [7].

In this paper, we look at the LDM process from a construction project perspective, assuming the contractor's early involvement in building information model (BIM)-based design. The authors of this paper are aware of LDM's technical and social aspects; however, this paper focuses mainly on the technical side of both LDM and information flow in the design process. This focus has been selected because there is lack of clarity on the technical side regarding how the different BIM concepts (such as level of detail) and LDM tools (such as Last Planner System) can be connected together.

The goal of this study is to define a process for design management by applying optimal lean tools, processes, and concepts for managing BIM-based design, focusing on the detailed design phase. The developed overall LDM process aims to prevent production delays caused by missing or incorrect design as well as to minimize the design team's rework caused by excessive levels of detail too early in the process [8]. The project's production strategy determines the milestones, which are the basis for scheduling the level of detail (LOD) of design information. Each LOD is based on information requirements for various end uses of design: permit process, procurement, prefabrication, or installation. Additionally, following lean principles, the LOD planning needs to be a pull-based process pulling from a location-based production schedule.

The paper first introduces the theoretical background on lean design management and LOD in BIM-based design. Then, we explain our design science research method for developing a new process. After that, current design management processes and challenges are analyzed, and a new process for tackling those challenges is presented. The process is then validated in a case project. The findings are discussed in light of previous literature, and finally we conclude by summarizing the theoretical contributions and implications for practice and suggest avenues for further research.

## 2. Background

### 2.1. Lean Design Management

Lean construction and lean thinking in design management have been studied for several decades [1,9]. Lean design management is a collection of lean methods, tools, and social processes that can be used to facilitate design [7]. Lean principles provide a structured means to improve the entire design system, which often consists of multiple specialized organizations and professionals. Lean design management reduces waste and improves customer value and information flow between organizations and project tasks [10].

The information in the design management process adds value when it is transparent and flowing between project parties [10]. A superb tool for improving information flow and transparency is the Dialogue Matrix (DM). DM promotes common understanding and facilitates dialogue, especially in meetings between project parties. The matrix identifies preconditions for design tasks and supports the pull logic. DM is structured to systematically record questions and answers from team members to other team members in the matrix system [11].

Another system for managing design as well as production is the Last Planner System (LPS). LPS is a well-known, collaborative, commitment-based, mainly social process, which integrates (1) *should*—setting milestones and strategies as well as specifying handoffs and identifying operational conflicts, (2) *can*—making work ready and replanning when needed, (3) *will*—making promises in weekly work planning, and (4) *did*—measuring success with PPC (plan percent complete) and acting on reasons for task failure [12]. The four levels of LPS scheduling are master schedule, reversed-phase schedule, look-ahead schedule, and weekly schedule.

LPS uses collaborative social methods in reverse phase scheduling (also called pull planning) to obtain task dependencies and commitments from project stakeholders [13], but it does not provide any guidance regarding those tasks should be in a BIM-based process. For example, it is not intuitively clear how LOD should be represented as LPS assignments on different scheduling levels. On the other hand, location-based methods, such as the location-based management system (LBMS) (e.g., [14]) and takt planning (e.g., [15,16]), have provided more structure for scheduling production by using LPS. In takt planning, takt refers to the regularity or intervals of tasks performed. Hopp and Spearman [17] defined Takt as the unit of time within which a product must be produced to match the rate at which that particular product is needed. In other words, it involves balancing the supply rate to match the demand rate. Originally, takt time was developed for manufacturing, but early results in construction production have been promising (e.g., [18]). However, it remains unclear how takt principles can be used to schedule design in a BIM-based process.

In summary, lean design management tools and processes each have their own strengths and optimal phases when they should be applied in a project. However, more knowledge is needed on how the use of those tools and processes should be combined in a single project and how the tools and processes should be connected to the BIM-based design process.

*2.2. Level of Detail in BIM-Based Design*

BIMForum, the US chapter of buildingSMART international, has promoted reference standards of LOD [19] for the construction industry. LOD has evolved from early concepts into a clear industry standard specification for communicating the content of BIM deliverables. Earlier approaches in LDM did not explicitly include the LOD of BIM models in connection with planning methods. LOD is a process in which building information models and the complexity of their components progress from the lowest level of conceptual representation to the highest level of detail based on the component's use; for example, for fabrication or installation needs [7,20,21]. However, determining an appropriate LOD for each element is only part of the solution, and so far, LOD has been considered in isolation with limited connection to design schedules. The total time spent in modeling increases vastly when going from one LOD to another [20]. If changes occur, the hours spent developing the model in too much detail ahead of the actual demand can be considered a waste.

Another system for managing detailed design is called location-based design management (LBDM). In LBDM, detailed design is done in production-determined clusters and is managed by location. LPS is used to manage design hand-offs. LPS pull scheduling is implemented so that milestones are formed by every location, and a location-based production schedule drives the design so that modelling and document production utilizes the same locations as construction and they are sequenced in the order of construction [7]. However, the method does not explicitly consider explicitly the LOD of the information. We argue that the combination of these two technical approaches, LOD and LBDM, could be a powerful way to enhance lean design management in a BIM-based project.

## 3. Method

The selected research method is design science research, and we attack a real-world problem [22], which was identified as the poor connection between production and design schedules. Design science involves a rigorous process of designing artifacts to solve problems. Design artifacts include operational concepts and practices, implementation methods, and instantiations [23]. In our case, the poor connection between the detailed design and production phases requires the construction of an operational LDM process. To become oriented in the current status of design management, the research team first conducted eight semistructured interviews with experienced construction professionals from several Finnish construction companies. The questions were prepared using findings from a literature review and aimed to find answers to the research questions. The interviews were conducted in the Finnish language from May 18 to July 18, and each of the interviews was recorded and transcribed verbatim. The average length of interviews was 50 minutes. Before each interview, the interviewees were briefed on the purpose of the study. After the transcription, an Excel spreadsheet was created in order to organize the topics and find similarities and discrepancies.

The developed solution artifact is a lean design management process, which was developed by combining earlier research and best practices related to BIM scheduling processes. Following Gregor and Jones's design theory components [24], the developed process description has the structure and functionality to implement lean theory in the context of detailed design. The process was validated in a focus group meeting. Focus group discussion is a sound method for collecting the ideas, views, and understandings of professionals with similar experiences from the same field on the subject at hand [25]. The focus group participants were invited from 13 Finnish design and construction companies, and each member of the focus group held a managerial position. The feedback and the general consensus of the focus group related to the design management process helped the authors

to validate and refine the technical and social parts of the process. Finally, the process was further validated by using it to plan the BIM design process of a case study project with the project team.

## 4. Interview Results from Finnish Construction Professionals

The interviews revealed the following details about design management practices in Finnish construction industry: (1) LPS is extensively used in construction companies, (2) LOD specifications are known at the end of project, but forecasting when those specifications are needed in the middle of the project is hard. (3) Designers are frustrated by the current process, where contractors demand too detailed design too early in the project. (4) Contractors are aware that they demand too much information too early in the project. (5) Designers question contractors' early design demands. (6) Design demands are generally divided into work packages for procurement reasons but there is no common understanding among project stakeholders of what information is required for each procurement package. (7) The outcome is delayed design and design which does not meet the requirements for procurement and production process.

## 5. LDM Overall Process

The technical system of the proposed LDM Overall process is based on Last Planner's four levels of scheduling. Different techniques and tools are used on different levels, and they link to the production schedule differently. The developed process is a combination of LOD, LPS, BIM, LBDM, and either LBMS or takt production scheduling. The novelty of the LDM Overall process is in scheduling the LOD of building information models using location-based methods and then using LOD to implement LPS. Table 1 presents the main components and sources of the LDM Overall process.

**Table 1.** Lean design management (LDM) Overall process and its main components.

| Scheduling Phase/Component | Description of Functionality | Applied Tools and Concepts |
|---|---|---|
| Master design schedule | Linking the production schedule to information needs and milestones based on project teams' decisions and production strategy | LPS + LOD (BIM) + LBMS/TAKT + LBDM |
| Phase design schedule | Collaborative pull plan meetings to identify information needs to accomplish current milestone | LPS + LOD (BIM) |
| Look-ahead schedule | Combination of look-ahead scheduling and DM for asking questions and documenting them | LPS + DM |
| Weekly schedule | Weekly meetings for the project team | LPS |
| Performance evaluation | Measuring success with plan percent complete (PPC) | LPS |

The developed LDM Overall process is a combination of LDM tools taken from the general LDM domain. The process is a guideline for applying the separate existing LDM tools together in a structured manner. However, since each project is unique in its design and delivery, the Overall process allows room for the project team to fine-tune the process to fit the requirements of the project.

### 5.1. Master Design Schedule

At this level, the design schedule is integrated with its equivalent counterpart—the master schedule of production. The master design schedule is based on controlling the LOD of the information and BIM models as well as linking the design schedule to production demands. Building permits, procurement, prefabrication, and installation are associated with individual information demands, which need to be defined at the beginning of the project. Some of the systems and information demands can be location-based, in which case their demand times are defined by the locations.

The LOD requirement is presented in LOD specification numbers as a function of time. The adjusted examples are based on the BIMForums LOD specifications [19]. LOD 350 contains information required by the installation, and that information must be ready before the first installation related to that particular system begins. LOD 325 is a coordinated and clash-free model. It contains information related to prefabrication, and it is scheduled considering the time needed for installation time and

fabrication prior to that. LOD 300 is a model in which the model elements have accurate geometry and their positions are accurately defined. LOD300 models can be used as a starting point for developing detailed models for prefabrication and installation purposes. Because the LOD specification is mostly focused on the details of the included geometry, we added a new level, LOD 290, where all the required elements exist with accurate geometries, but the model is not necessarily fully coordinated, and the exact positions of the elements are still subject to change. In LOD 290, all the necessary model elements and their accurate geometries are in the model, but the elements are not yet in their final positions. LOD 290 models can be used for procurement or quantity take-off purposes but cannot be used to guide installation or prefabrication efforts. LOD 200 contains preliminary designs. Each LOD and the documentation related to it for each system is attached to the master design schedule. The idea is to schedule the demand for information and coordinate that demand to the construction process.

In order to tie this idea to a location-based schedule (e.g., takt or LBMS schedule), the following issues should be defined: (1) what information is demanded and when, (2) what information can be developed by locations, and (3) what information should be developed by systems. Here, the information is mainly referring to BIM-based information; however, information may also be presented in forms other than BIM, such as text (e.g., specifications). Every time the level of detail increases from one LOD to another, a responsible organization is needed to perform the required design work. The design work often includes several design assignments, which can be managed with LPS.

Figure 1 presents a practical example of linking design tasks to production tasks that are location-based in a takt or LBMS schedule. This example involves the design and manufacturing of structural precast concrete elements and installing them from floors two to six. It takes five days to install all the structural elements for each floor before proceeding to the next floor. The information needs are different in each phase. For example, the installer requires more detailed design in the installation phase than the precast company does when making a bid in the procurement phase. The white area on the left in the figure, just before the yellow area (LOD 290), shows the preplanning phase. Here, that phase is marked as "Task 1" on the lower part of the schedule. Procurement-level information for all the elements of floors two to six are developed in the yellow area. Both of these areas are designed as system-based (the LOD milestone occurs at the same time for each milestone). Procurement-level information would be, for example, detailed model-type elements at LOD 350 of each element type—wall, pillar, beam, and hollow core slab—as well as volume and surface area information from all the BIM model elements at LOD 300. This information would be enough for a precast plant and an installation company to make a bid for the work, assuming standard connection details. Design for prefabrication (LOD 350, blue area) can continue in a location-based manner, and the location-based design deliverables can become a part of the takt train at this point. Responsibility could also switch at this time, and the selected precast company could become responsible for taking the models to a prefabrication level of detail. By the time installation starts in the first red square on the second floor in Section 1, the design must contain all necessary information for installation (LOD 400). Each LOD milestone is used as a phase-scheduling milestone, and a pull-scheduling LPS process is used to plan the tasks that are necessary to increase the LOD to the desired level. The lower part of the schedule contains the design tasks resulting from the LPS process. They are not necessarily location-based but have dependencies with the location-based design tasks (red arrow lines indicate those dependencies).

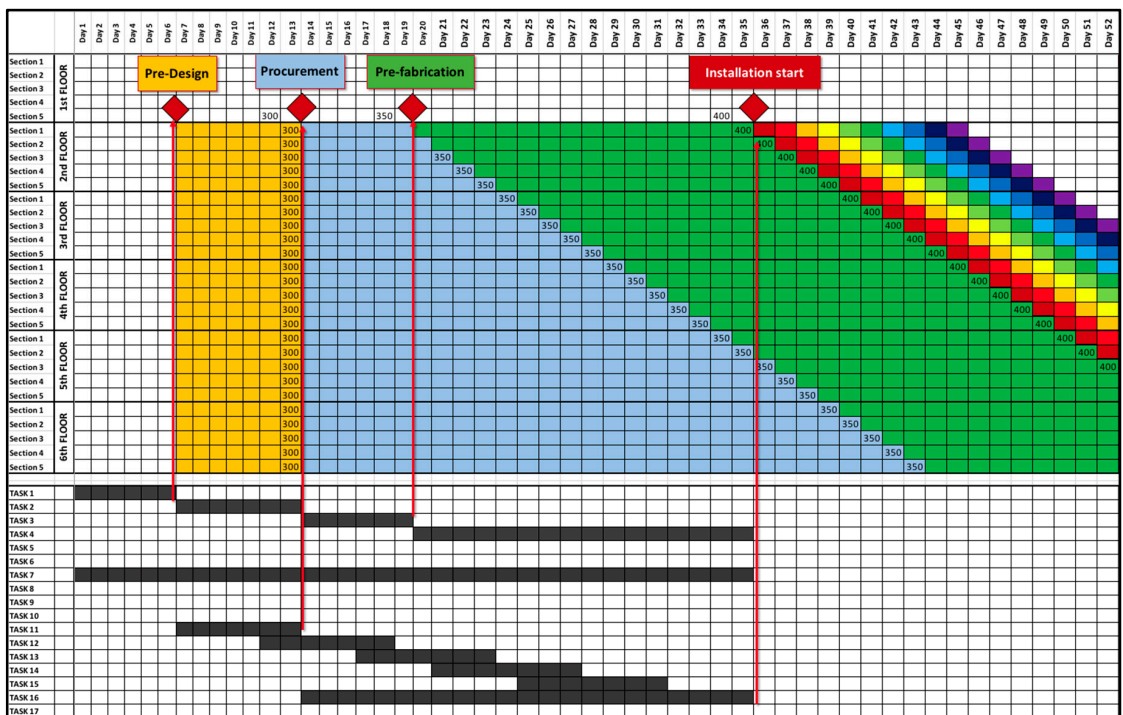

**Figure 1.** Linking the design LODs with the production takt time. Vertical axis presents locations and horizontal axis presents time.

## 5.2. Phase Design Schedule

The basis of phase design schedule is performed as a standard LPS reversed phase schedule. LOD demands from the master design schedule act as milestones. These milestones are made either for one system or several interconnected systems, such as the structural framework, interior finishes, and HVAC coordinating. The reversed phase schedule is performed by starting to pull from the required LOD for the selected scope. The project team, which produces the phase design schedule, consists of a general contractor, trade partners, and the designers.

First, the project team needs confirmation of mutual agreement of the final result of the phase (e.g., [26]). The final result embodies what information demands are needed at the end of the phase. Second, planning starts from the end and moves towards the beginning. When placing a Post-it note, the person requests information from the other project party. The focus is to progress based on information demands rather than listing each party's design tasks. The objective of this procedure is to delete unnecessary tasks. The results from these unnecessary tasks are not demanded by anyone.

Design Structure Matrix (DSM) is a complementary tool to resolve task dependencies. DSM is a tool that defines tasks, their relations, and information needed from other tasks, and from that information an optimal task sequence is indicated in the matrix [27,28]. Although DSM and LPS have overlapping properties, and while DSM is a more technical tool lacking the social aspects of LPS, the use of both tools would add to information flow inside the project.

The project team needs to resolve whether each location where takt work starts counts as a milestone. This resolution depends on the decision made on the master design schedule level—that is, what parts of the design are location-based. Similar to production, the phase schedules of each similar milestone are likely to be very similar, so pull scheduling could focus on just one milestone, and then the team could determine any differences between locations that would necessitate variations from the template phase schedule.

## 5.3. Look-Ahead Design Schedule

The basis of this phase is standard LPS look-ahead scheduling. The overall process adds a more structured way of tackling emerging obstacles by combining the use of DM into this phase. All the questions asked by project party are then documented using DM. Answers to those questions will turn into tasks for the answering party. Generally, questions are equivalent to obstacles in LPS backlogs.

## 5.4. LDM Overall Process Validation

The authors of this paper presented the LDM Overall process in a focus group meeting for validation. The focus group shared a couple of concerns. The first was how the Overall process would operate with different project contract types in which the LOD requirements for different phases are very different (e.g., fixed price requires an almost complete design at procurement). The solution would be to have a project contract type-based specification of LOD requirements at each step. The second concern was how to choose the design tasks that should be scheduled based on location and how to choose those that should be dealt with at a system level. This is an open question, which can be addressed in future research. However, one promising method would be to apply the concept of product modularity or an open building approach to distinguish between systems and their designs that require a whole building approach and sub-systems in locations that can be designed independently from others, which are therefore appropriate for location-based design schedule. In general, all the focus group participants were ready to accept the process and its concepts and liked the schedule's simplicity and its visual performance. The process conceptually answered the requirements and identified challenges in the current practices. For further validation, the authors decided to apply the process in a case study.

## 6. Case Validation

In the second phase of the study, the process was validated by applying the process and its components in a case project based on a workshop with the project team. The purpose of the additional validation was to evaluate the feasibility of using the developed LDM process to enhance the case project's current daily design management routines. The chosen case was an underground metro project located in southern Finland. The case project consists of five new stations, a depot, and seven kilometers of rail line as well as several shafts for pressure equalization, ventilation, and smoke extraction.

The preliminary design phase was completed at the end of 2016, followed by a detailed design phase. The detail design teams—rail engineering, architecture, structural, rock engineering, HVAC, fire protection, and electrical engineering—utilized BIM technology to deliver designs for the project. Due to the sheer magnitude of the project, several design offices participated in the project. Hence, the use of a big room was required to enhance information flow and to ensure collaboration among the different design firms.

## 6.1. Design Management Process and Challenges in Case Project

Before the new LOD process was considered, the case project was using mostly a traditional approach in its design management process, where problems were solved after encountering them, and handoffs and the need for their input were unclear. However, some lean tools and methods had been incorporated into the case project's process. From March 2017 until June 2018, the project employed Scrum as a tool to manage structural engineering and design among several design firms. Scrum is a method, originating from software development, in which the pull principle is highlighted to manage production of design deliverables by limiting the work-in-progress [29]. The case project implemented LPS at later parts of the detailed design phase. The purpose of LPS was to manage a reasonable workload and to generate necessary pull for other design parties—architecture, HVAC, and electrical. However, LPS was not used systematically and was only used in two out of the five stations. The software used to implement Scrum and to manage backlogs and sprints in structural

engineering was called JIRA. The software used in the LPS sessions was called TRELLO, and it was used similarly to JIRA in structural engineering. Both programs were provided by the same Australian company (Atlassian).

While tracking roughly 92,000 hours of structural design work from early 2017 to mid 2018, JIRA provided biweekly PPC for the work performed. PPC was also further used to improve the current design management process. In addition, JIRA was used for reporting the structural engineering status as well as for forecasting. The average PPC in 2018 was approximately 56%, and that result correlates with the typical poor productivity of the construction industry [30]. The required information content of the designs at milestone—mid-2018, at the end of detailed design phase—was 100% of the LOD 300/350 documents. However, contractors and trade partners were not involved when document handovers were inspected. The principal designers were inspecting the quality and information integrity of the client's design documents as well the documentation produced by the contractor. Figure 2 represents the inspection process duration in one station, and it shows how many days each inspection took. The owner set the lead-time target for the contractors' document inspection to ten working days. Although this target was reached on average (mean 12.0 calendar days), there was considerable variation. Figure 2 clearly shows alarming spikes in the inspection process. A fivefold lead time for construction document inspection would be detrimental to construction production if takt production was implemented by the contractor. One of the reasons for delayed inspection was the large batch size of the inspected documentation. Another reason was the lack of communication between the contractor and principal designer.

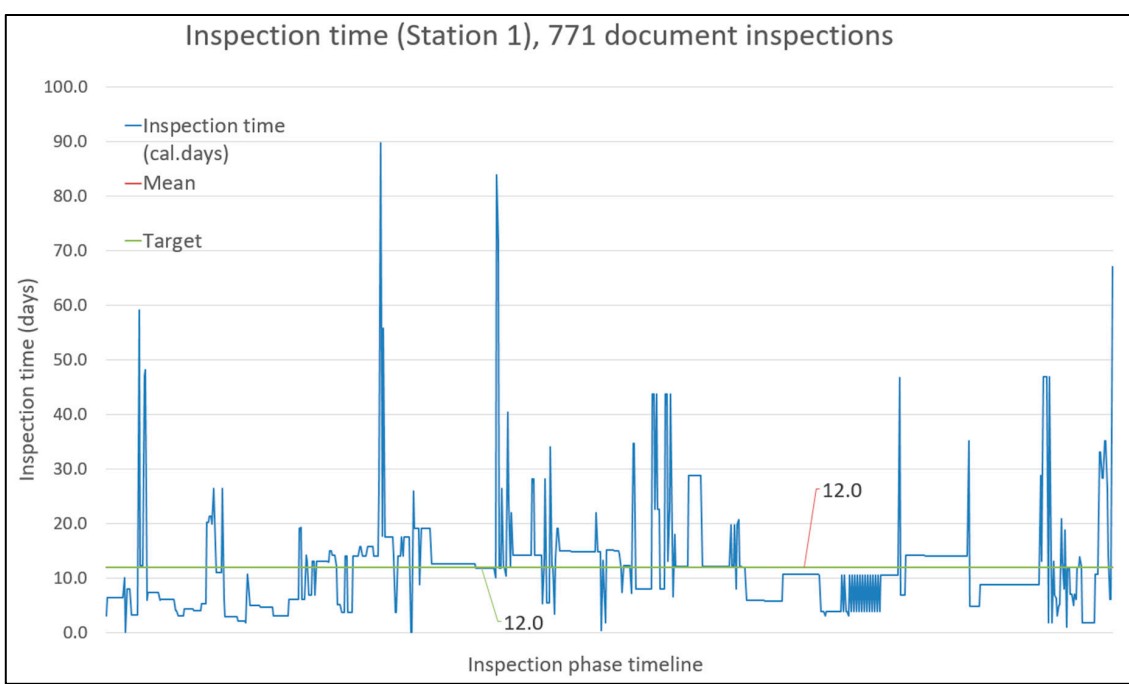

**Figure 2.** Representation of case project's construction document inspection time in the inspection process.

The project team expressed their concerns regarding takt time production and how to ensure the delivery of inspected construction documents for the beginning of each takt. The team also wanted a process for the possible event of incorrect construction documentation while in the middle of takt (e.g., so-called disaster protocol related to construction documents).

### 6.2. Proposed Means for Implementing the Overall Process in the Case Project

The authors and the project team agreed to develop an adjusted design management process to support the takt production of HVAC and electrical installations. Figure 3 compares the main components of the Overall process and the case project's level of implementation in each of those components. The level of implementation was determined with the design manager of the case project. From the low end of the scale, very low means that the project team has no experience with that component. Meanwhile, very high means that the project team is familiar with the component (e.g., JIRA and TRELLO were used in a previous phase of the project). JIRA and TRELLO have overlapping properties with DM, and the project team already had established working practices for using the solutions. Therefore, rather than using DM, JIRA and TRELLO were proposed to replace that component of the overall process.

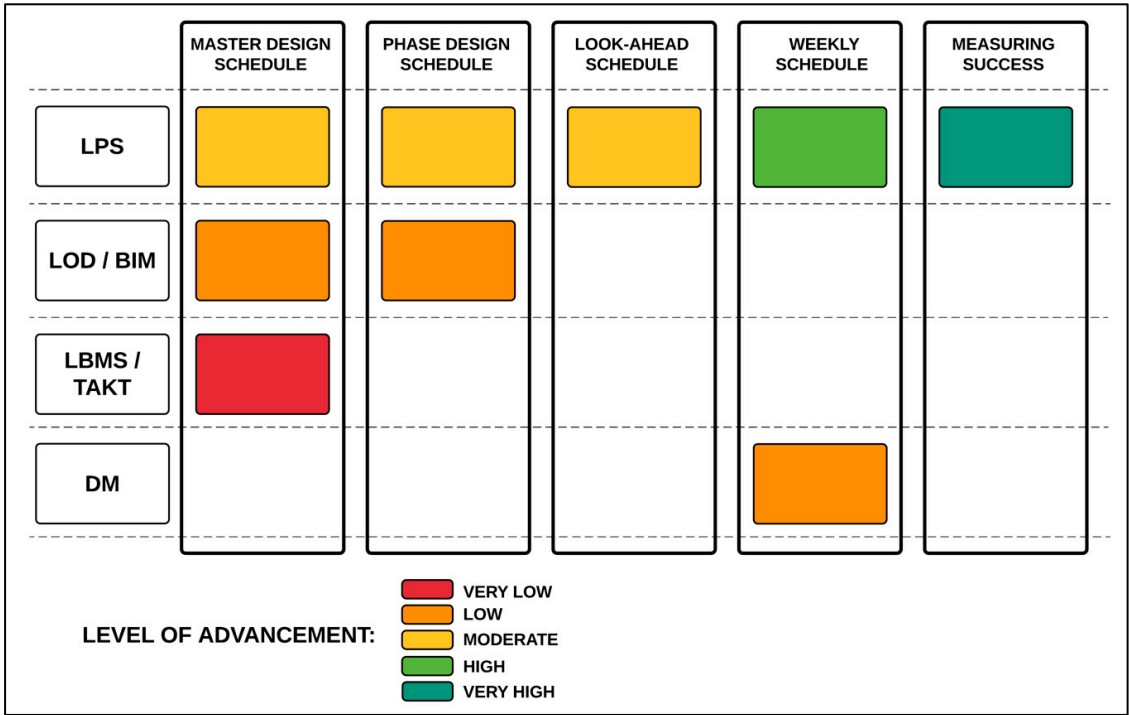

**Figure 3.** Case project's experience of each of the Overall process components.

To address the project team's concerns related to the construction document inspection protocol, the authors developed a two-phase inspection process for the HVAC and electrical installation documents. The first inspection is due four weeks before the first work of a takt-area commences in the particular HVAC-system level. The inspection workshop is a collaborative big-room session in which the HVAC designer and trade partner participate. The trade partner is contractually obliged to participate in the detailed design phase. The first inspection contains the final clash detection of HVAC ducts in a larger system-level area, which covers many takt-areas. Additionally, a generic suspension and mounting plan is laid out in collaboration with the installer and designer. The second inspection is due one week before the first takt-area on that system level commences. The second inspection workshop is another collaborative big-room session with the same participants as the first inspection workshop. In the second workshop, the installer confirms the exact suspension and mounting plan and detailed measurements for the installations for each takt-area at the system level. Figure 4 shows the systematic inspection process implemented in the takt schedule. Here, a takt time of one week is proposed because HVAC installations form the majority of total work to be performed. Moreover, the layout of the case project's metro station is challenging in that the installations are mostly nonrepetitive.

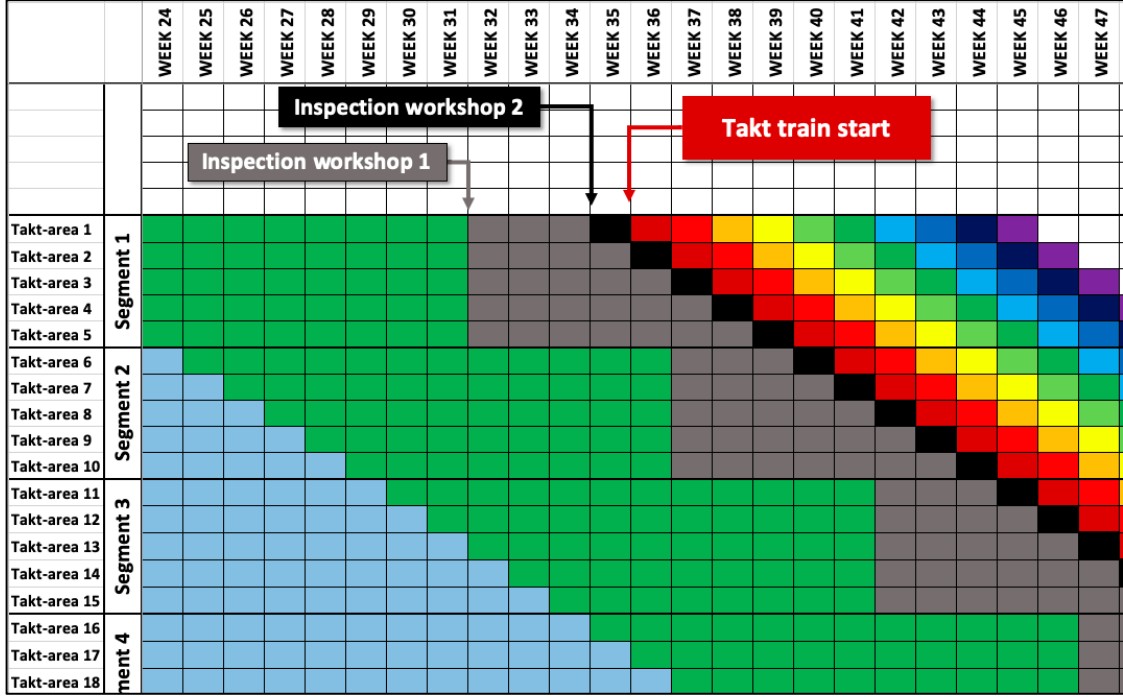

**Figure 4.** Two phases of inspections for HVAC construction documents.

Related to possible issues found in the second inspection workshops, the authors propose two policies for reacting to those issues. The first option is a field engineering arrangement, where the HVAC designer is on-site daily and can react rapidly and update the plan and construction documents. This is the preferred option since issues can be solved face to face with the project team. The second option for a reaction policy is an on-call service where a designated HVAC designer prioritizes sudden design needs. Here, any issues are solved using virtual methods and tools; hence, this option is the second choice for problem solving.

Since the case project's team already has moderate experience in using LPS and measuring PPC, the implementation of the lacking components of the Overall process should be reasonably elementary and should not add much additional burden on the project teams' resources. The main focus is on adjusting the design work and inspection process to align with the takt schedule. Additionally, the scope of big-room working needs to be increased to include defining the information needs for each takt-area and the required LOD.

In summary, based on design management challenges of the case project so far, the feedback from the workshop, and the created implementation plan, the Overall process could benefit the case project's design management routines greatly. In addition, the Overall process could lower the risk of undelivered construction documents, which would ultimately hinder the contributions of the takt time schedule.

## 7. Discussion

Good best practices for lean design management were found from earlier research, interviews, and case projects, but each one only solved a part of the problem. This research contributes to the existing knowledge on lean design management by suggesting a process model for specifying how different lean tools and processes could be used in different phases of the project to align the schedule of detailed design tasks with construction activities. By combining information from various sources, we were able to define a location-based design management process using the concept of LOD, which can be integrated with location-based methods and LPS. The LOD definition must start from the end-user's requirements for information in each stage of the construction project.

For example, to start procurement, we do not need final positions of each building information model element; rather, we need a way to estimate the quantity of work and risks to come up with the price estimate. For prefabrication, we need a fully coordinated building information model to ensure that the prefabricated parts fit without field rework. For installation, the BIM-based design must include all the necessary details required by the workers. The design for prefabrication and installation can be location-based and can be pulled by the production schedule. The fact that LOD is currently defined according to the geometry of the BIM elements limits the full potential of the LDM Overall process. To complement the LOD specification, it is important to determine what information is required at each stage and which parts of the models can be scheduled by location. When applying LOD in a project, the project team needs to collaboratively agree on how to leverage the BIMForums LOD standards and how to adjust that standard based on the project's individual information needs. Moreover, the project needs may vary due to different contract forms and different responsibilities of actors participating in the design, prefabrication, and construction processes.

The suggested overall design schedule is partially system-based and partially location-based. The LPS is used to elaborate the design schedule and get to the level of commitment of individual designers to verify the starting data requirements and required decisions for each design task. In this way, the process involves several stakeholders: the owner, architect, engineers, and the general contractor as well as trade contractors.

## 8. Conclusions

The proposed LDM process can solve most of the design management challenges identified from the interviews of Finnish construction professionals. Since LPS is already used extensively in Finland, it would be fruitful to intensify design management to collaboratively manage the timing and the level of details and designs in a project. If every project party would commit to their task, there would be no need for the contractor to demand an overly detailed design early in the project. The LDM Overall process steers the project stakeholders to thoroughly—and in a structured way—discuss upfront the information needs and their interdependency with building information models related to each procurement package, thus increasing common understanding among the project team. The Overall method also has the potential to reduce waste in the current BIM modelling process by scheduling accurate LOD of building information models based on pull from production demands. By reducing waste in the modelling process, the Overall process contributes to streamlining the entire project design process, thereby improving the productivity of the project. Further, shifting the focus from merely optimizing the design process to optimizing the whole production system by pulling from the construction production process will lower the risk of overrunning the planned lead time of the project. When all project parties commit to the scheduled LOD-based milestones, there is no need for modelers to create overly detailed models based on assumptions, which could result in remodeling when changes occur.

The LDM Overall process was cocreated and validated in focus group meeting with engineering and construction companies. Further validation was performed by adapting the proposed process to the circumstances of the case project. In this way, the research was able to tackle a real-world problem with a solution artifact, which was validated by industry professionals and one case project. However, a limitation of this research is that it did not demonstrate actual outcomes of the developed process. Therefore, in future research, we will perform a case intervention and document the results. Additional case studies are needed to test and further develop the Overall model in real-world settings.

**Author Contributions:** Conceptualization, P.U., O.S. and H.O.; methodology, P.U., O.S., A.P. and H.O.; validation, P.U., O.S., and E.L.; formal analysis, P.U.; investigation, P.U.; data curation, P.U. and E.L.; writing—original draft preparation, P.U. and O.S.; writing—review and editing, P.U., O.S., E.L., A.P. and H.O.; visualization, P.U.; supervision, O.S. and A.P.; project administration, O.S.; funding acquisition, O.S. and A.P.

**Funding:** This research and the APC was funded by Building 2030 consortium.

**Conflicts of Interest:** The authors declare no conflict of interest. The funders had no role in the design of the study; in the collection, analyses, or interpretation of data; in the writing of the manuscript, or in the decision to publish the results.

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
