# Peer review of "Applying Level of Detail in a BIM-Based Project: An Overall Process for Lean Design Management"

_buildings, doi:10.3390/buildings9050109_

Reviewer 1 Report

This is an interesting theme which has gained attention of many authors. Paper is well structured and gives an overview of other research in the area. The structure is logical and easy to follow.

Author Response

We thank the Reviewer for comments and feedback.

We have used our time in making sure that the text is grammatically correct and the text flows more fluently.

Reviewer 2 Report

1. The authors claim that “the goal of this study is to define a process for design …”. However, the process is not clear. Is the process equal to the LDM process?

2. The title for section 4 is not fitted to the content.

3. The LOD defined in the United States is just a concept, not a clear specification. How to integrate a concept and a real tool (LDM) requires more illustrations. Furthermore, there is no LOD 290 in details in BIMForum. It is necessary to give more definition even it is just a concept.

4. The advantages and limitations of this research should be highlighted.

5. It is suggested to separate the last section (the discussion and conclusions) into two independent sections.

6. The LOD is a concept only. How to use the concept for a user is a difficult task. It is necessary to illustrate the use case.

Author Response

We thank the Reviewer for good comments and feedback. All the edits are marked in red font in the revised manuscript. We believe that the manuscript is now more refined after Reviewers comments have been incorporated into text. Text is now grammatically correct and the text flow is more fluent.

In general:

Language of the paper has now been edited by a native English speaking person provided by an outside service.

In section 3, Methods, we have added more text to method selection and added more detail to how the interview research was conducted.

The results of the research is the developed artifact (design management process) and we have added more explanations in section 5.

Point 1: The authors claim that “the goal of this study is to define a process for design …”. However, the process is not clear. Is the process equal to the LDM process?

Response 1: We have renamed the headings in Table 1 to better describe the components of the developed process. Also, we have indicated that the developed process is a combination of current existing LDM tools in Page 4 (lines 163-166). We mean that LDM is a general domain of design management and the developed process is one tool in that domain.

Point 2: The title for section 4 is not fitted to the content.

Response 2: Title has now been renamed to better describe the content of the section 4.

Point 3: The LOD defined in the United States is just a concept, not a clear specification. How to integrate a concept and a real tool (LDM) requires more illustrations. Furthermore, there is no LOD 290 in details in BIMForum. It is necessary to give more definition even it is just a concept.

Response 3: We have pointed out that BIMForum’s latest LOD Specification are not anymore at concept stage (Page 3, lines 98-99 and Page 11, lines 390-397). We argue that they are, a well-defined and ready to use BIM LOD specifications, with extensive illustrations and descriptions of sufficient coverage over various BIM elements.

Also, we have added an explanation related to LOD 290 in page 5 (lines 181-188).

Point 4: The advantages and limitations of this research should be highlighted.

Response 4: We sharpened the process advantages in page 11, lines 414-418 and lines 423-424. We also added limitation of the research in lines 425-426.

Point 5: It is suggested to separate the last section (the discussion and conclusions) into two independent sections.

Response 5: We have separated the last section into two independent section.

Point 6: The LOD is a concept only. How to use the concept for a user is a difficult task. It is necessary to illustrate the use case.

Response 6: This is an important point. We have added more details to clarify our example use-case from Figure 1. Additions are added within lines 200-218 in Page 5.

Reviewer 3 Report

The article is of average quality, as the topic has been researched before. Nevertheless, it is important for its field of research. The language used could be improved.

Author Response

We thank the Reviewer for comments and feedback.

The manuscript has now been edited by a native English speaking person provided by an outside editing service. Text is now grammatically corrected and more text flow is more fluent.

Round  2

Reviewer 2 Report

All comments in the first round have been addressed well.

No other comments.